

# Genome-wide survey of single-nucleotide polymorphisms reveals fine-scale population structure and signs of selection in the threatened Caribbean elkhorn coral, *Acropora palmata*

Meghann K. Devlin-Durante[*] and Iliana B. Baums[*]

Department of Biology, Pennsylvania State University, University Park, PA, United States of America
[*] These authors contributed equally to this work.

## ABSTRACT

The advent of next-generation sequencing tools has made it possible to conduct fine-scale surveys of population differentiation and genome-wide scans for signatures of selection in non-model organisms. Such surveys are of particular importance in sharply declining coral species, since knowledge of population boundaries and signs of local adaptation can inform restoration and conservation efforts. Here, we use genome-wide surveys of single-nucleotide polymorphisms in the threatened Caribbean elkhorn coral, *Acropora palmata*, to reveal fine-scale population structure and infer the major barrier to gene flow that separates the eastern and western Caribbean populations between the Bahamas and Puerto Rico. The exact location of this break had been subject to discussion because two previous studies based on microsatellite data had come to differing conclusions. We investigate this contradiction by analyzing an extended set of 11 microsatellite markers including the five previously employed and discovered that one of the original microsatellite loci is apparently under selection. Exclusion of this locus reconciles the results from the SNP and the microsatellite datasets. Scans for outlier loci in the SNP data detected 13 candidate loci under positive selection, however there was no correlation between available environmental parameters and genetic distance. Together, these results suggest that reef restoration efforts should use local sources and utilize existing functional variation among geographic regions in *ex situ* crossing experiments to improve stress resistance of this species.

Corresponding author
Iliana B. Baums, baums@psu.edu

## INTRODUCTION

There is an ongoing debate about the importance of local recruitment and barriers to gene flow in marine species. Many marine species reproduce via planktonic larvae and strong ocean currents have the potential to carry propagules over long distances. However, a high degree of self-recruitment has been found in a range of species with planktonic larval duration being a poor predictor of genetic structure (*Selkoe & Toonen, 2011*). The

development of cheap genome-scale genotyping is poised to open a new chapter in this discussion (*Peterson et al., 2012*; *Toonen et al., 2013*; *Wang et al., 2012*). American eels for example show panmixia in their central breeding ground in the North Atlantic but single nucleotide polymorphism (SNP) genotyping of adults along the Eastern seaboard revealed local differentiation (*Gagnaire et al., 2012*). Thus, a well-mixed pool of larvae sorted into environmental niches and so resulted in a structured adult population.

SNPs are ubiquitous throughout the genome, located in coding and non-coding regions, and each locus has a maximum of four alleles (the four bases). This is in contrast to microsatellite loci that consist of tandem repeats, in which allelic variation is determined by the number of tandem repeats and thus can be large. The limited number of alleles at each SNP locus requires a larger number of loci to be assayed to achieve the same power of detecting population genetic structure as a panel of microsatellite loci (*Morin, Martien & Taylor, 2009*; *Ryman et al., 2006*). The advent of reduced representation sequencing methods have made it possible to develop and assay a large number of SNP loci at a reasonable cost (*Altshuler et al., 2000*; *Hoffberg et al., 2016*). Recently, Genotyping by Sequencing (GBS) data including 4,764 SNPs in *A cervicornis* identified population structure within the Florida Reef tract (*Willing, Dreyer & Van Oosterhout, 2012*) where microsatellite markers did not (*Baums et al., 2010*). Other flavors of reduced representation sequencing methods (*Drury et al., 2016*; *Toonen et al., 2013*; *Wang et al., 2012*) have yielded information on population structure, and genetic diversity in reef building corals (*Drury et al., 2016*; *Howells et al., 2016*).

Genome-scale genotyping can provide insights into genetic diversity within functional regions of the genome that may be under selection (those genomic regions that code for proteins or regulate transcription of genes). These regions are not commonly surveyed even though they are of interest to conservation managers who want to understand how much capacity there is in a species to adapt to changing conditions (*Becks et al., 2010*). Statistical methods have been developed that allow scanning of SNP loci for signatures of selection. Despite the risk of generating false positive results (*Vilas, PÉRez-Figueroa & Caballero, 2012*), these methods yield candidate loci that should be substantiated by further testing (*Renaut et al., 2011*; *Sork et al., 2016*). The same methods can be used to scan microsatellite loci for signatures of selection (*Nielsen, Hansen & Meldrup, 2006*; *Vasemägi, Nilsson & Primmer, 2005*), however, power is often limited by the small number of assayed loci.

*Acropora palmata* is one of a few Caribbean coral species whose population genetic structure has been thoroughly investigated on local and range-wide scales (*Baums, Devlin-Durante & LaJeunesse, 2014b*; *Baums, Miller & Hellberg, 2005b*; *Baums, Miller & Hellberg, 2006a*). A range-wide survey of *A. palmata* population genetic structure using five coral specific polymorphic microsatellite markers showed that *A. palmata* stands are structured into two long-separated populations (*Baums, Hughes & Hellberg, 2005a*). While most reefs are self-recruiting, *A. palmata* stands are not inbred and harbor high genetic diversity at these microsatellite loci (*Baums, Miller & Hellberg, 2005b*). Bio-physical modeling identified a transient feature in the Mona Passage important in restricting present-day gene flow between the eastern and western population (*Baums, Paris & Cherubin, 2006b*). However, it is unclear whether the eastern and western populations differentiated initially

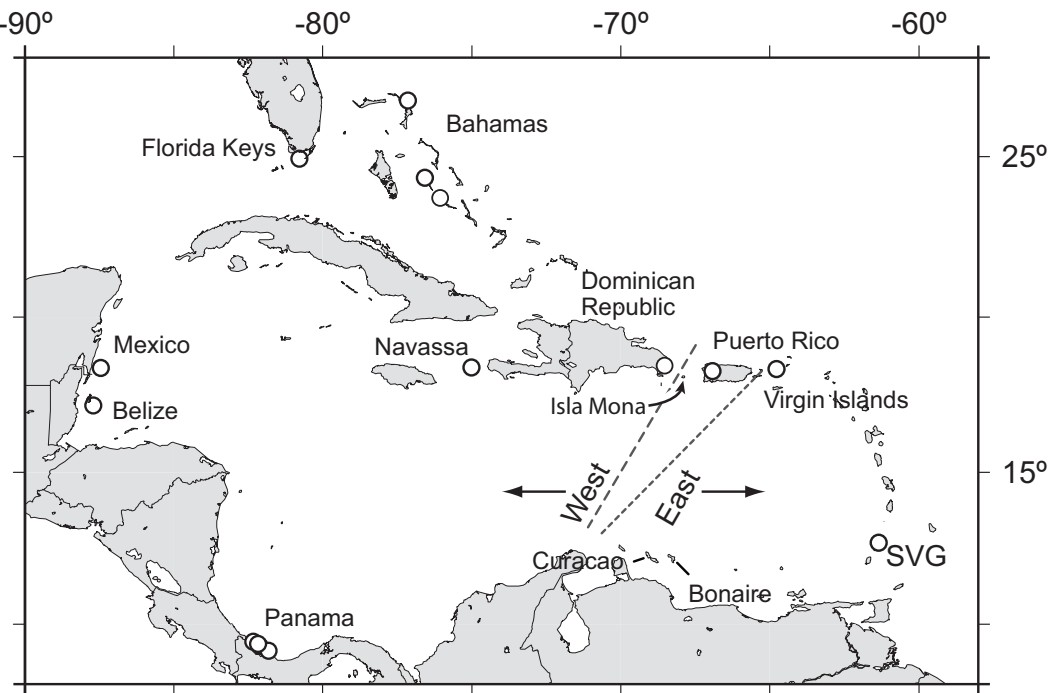

**Figure 1** *Acropora palmata* **samples were obtained from throughout the Caribbean and north-west Atlantic range.** Previous studies found a genetic break between the western and the eastern Caribbean but disagreed on the assignment of Puerto Rico to the western (long dashed line) or the eastern (short dashed line) population.

due to selection. Subsequent denser sampling of *A. palmata* along the Antilles Island Arc raised the possibility of a hybrid zone across Puerto Rico rather than a clear-cut break between the eastern and western Caribbean at the Mona Passage (Fig. 1, *Mège et al., 2014*).

We sought to refine the location of the east–west population divide and test for the presence of finer scale population differentiation in *A. palmata* by developing a large number of SNP markers. We assayed genome-wide SNPs in archived samples from two geographic regions in the western *A. palmata* population (Bahamas and Florida) and two geographic regions in the eastern population (Puerto Rico and the US Virgin Islands (USVI)). We then compared the results to population structure derived from eleven microsatellite loci. We further aimed to produce a more comprehensive estimate of genetic diversity across the genome using SNPs and screened loci for signatures of selection.

## MATERIALS & METHODS

### Sample collection

Colonies of *A. palmata* were collected between 2002 and 2010 and previously genotyped (*Baums, Devlin-Durante & LaJeunesse, 2014b*; *Baums, Miller & Hellberg, 2005b*). Unique genets were selected from our database for a total of 24 samples from each of four geographic regions; the Bahamas, Florida, Puerto Rico and the US Virgin Islands (USVI). The goal was to have eight samples from three different reefs within each geographic region,

however this was not always possible either due to small sample sizes from a particular reef or low clonal diversity of a reef. In those cases, we selected additional unique genets from nearby reefs. See Table 1A for detailed sample information.

We used an extended set of samples to compare the population genetic structure ascertained via microsatellite genotyping to the SNP results. This extended set of samples included 260 samples from six geographic regions; Belize, Florida, Puerto Rico, the USVI, and Curacao (Table 1B). Note that not all SNP-genotyped samples were included in the microsatellite dataset.

## Library preparation

Coral tissue samples were extracted from ethanol preserved samples using DNeasy Blood & Tissue Kit (QIAGEN, Hilden, Germany) with the following modifications. Time of incubation in the extraction buffer was increased to 16–20 h and two 100 µl elutions were performed, the second of which was kept for library production as this fraction contained the high molecular weight DNA. Extracted DNA was then treated with 0.01 mg of RNase A (10 mg/ml; Amresco Solon, Solon, OH, USA). Extraction concentrations ranging from 500 ng to 6 µg were double-digested with 10 units of each of the restriction enzymes MluCI (´AATT) and NlaIII (CATG´) (New England Biolabs, Ipswich, MA) following the protocol described by *Peterson et al. (2012)*. Digestions were purified using 1.5× Ampure beads (Beckman Coulter Inc, Brea, CA, USA) and quantified on a Qubit® fluorometer (Life Technologies, Carlsbad, CA, USA). Digested DNA was standardized to 100 ng for each sample before adaptor ligation. Samples were identified with eight 6-bp indices on the NlaIII (rare-cutter) P1 adapter (Table S1). Samples were pooled into 12 libraries and then size selected in the range of 200–800 bp on a Pippin-Prep (Sage Science, Beverly, MA, USA). Next, Illumina flow-cell annealing sequences, unique multiplexing indices and sequencing primer annealing regions were added through PCR amplification to the MluCL cut end (See *Peterson et al., 2012*, Protocol S1, Fig. 1). The libraries were enriched with 12 amplification cycles in four separate PCR reactions for each library containing 10 µl of Phusion High-Fidelity PCR Master Mix with HF Buffer (New England Biolabs, Ipswich, MA, USA), 2 µl of each amplification primer, 1 µl of library DNA and 5 µl of water (total 20 µl). Samples were pooled into four libraries each containing 24 samples (Table 2, Table S1). Each library was sequenced on one lane of Illumina HiSeq 2000 sequencer (paired-end, 2 ×150 bp) at the Pennsylvania State Genomics Core Facility. There were two libraries sequenced on each chip. See *Peterson et al. (2012) Supplementary Information* for a detailed protocol. Radseq methods have been used successfully in scleractinian corals (*Combosch & Vollmer, 2015*; *Dimond, Gamblewood & Roberts, 2017*; *Forsman et al., 2017*) and other marine invertebrates (*Lal et al., 2016*; *Reitzel et al., 2013*).

## Raw sequence filtering

Raw sequence reads were filtered using the process_radtags in the pipeline Stacks 1.21 (*Catchen et al., 2013*; *Catchen et al., 2011*). Barcodes and the RAD-Tag cut sites were identified to de-multiplex the pooled data into individual samples (Table S2). Reads were discarded that had low quality (with an average raw phred score <10 within a 15-base pair
**Table 1** *Acropora palmata* **colonies included in the SNP (A) and microsatellite (B) analyses.**
Samples were obtained from 3–6 (A) or more (B) reefs in four (A) and six (B) geographic regions in
the Caribbean/north-west Atlantic. Given are latitude and longitude in decimal degrees (WGS84).

| Region | Reef | Count of samples | Latitude | Longitude |
|---|---|---|---|---|
| **(A)** | | | | |
| Florida | Sand Island | 6 | 25.018093 | −80.368472 |
| | French | 8 | 25.03393 | −80.34941 |
| | Little Grecian | 1 | 25.118433 | −80.31715 |
| | Horseshoe | 1 | 25.139467 | −80.29435 |
| | Elbow | 8 | 25.143628 | −80.257927 |
| Bahamas | Little Ragged Island | 1 | 22.15375 | −75.687208 |
| | Adelaine Cay | 8 | 22.173372 | −75.703016 |
| | Elkhorn Cay | 2 | 22.328253 | −75.783228 |
| | Johnson Cay | 3 | 22.33312 | −75.77892 |
| | Nairn Cay | 8 | 22.35199 | −75.79612 |
| | Middle Beach | 2 | 23.781199 | −76.10391 |
| Puerto Rico | San Cristobal | 8 | 17.56493 | −67.04515 |
| | Rincon | 6 | 18.21007 | −67.15849 |
| | Tres Palmas | 2 | 18.350133 | −67.266333 |
| | La Cordillera | 8 | 18.368522 | −65.571678 |
| US Virgin Islands | Tague Bay | 8 | 17.763867 | −64.613397 |
| | Hawksnest Bay | 8 | 18.347183 | −64.780775 |
| | Johnsons Reef | 8 | 18.361733 | −64.7743 |
| **Grand total** | | 96 | | |
| **(B)** | | | | |
| *Florida* | Horseshoe | 1 | 25.1395 | −80.294 |
| | Little Grecian | 1 | 25.1184 | −80.317 |
| | Sand Island | 6 | 25.0179 | −80.369 |
| | Western Sambo | 6 | 24.4799 | −81.719 |
| | Rock Key | 4 | 24.456 | −81.86 |
| | Dry Tortugas | 1 | 24.6209 | −82.868 |
| | Marker 3 | 1 | 25.3733 | −80.16 |
| | Boomerang Reef | 1 | 25.3525 | −80.179 |
| | Carysfort | 4 | 25.2219 | −80.211 |
| | Great Iguana | 19 | 26.7075 | −77.154 |
| | Middle Beach | 2 | 23.7812 | −76.104 |
| | Charlies Beach | 1 | 23.7808 | −76.104 |
| | Black Bouy | 1 | 23.8022 | −76.146 |
| | Bock Cay | 1 | 23.8075 | −76.16 |
| | Little Darby | 2 | 23.8474 | −76.209 |
| | Rocky Dundas | 1 | 24.2788 | −76.539 |

**Table 1** (*continued*)

| Region | Reef | Count of samples | Latitude | Longitude |
|---|---|---|---|---|
| *Bahamas* | Halls Pond | 2 | 24.3539 | −76.57 |
| | LSI | 3 | 23.7691 | −76.096 |
| | Little Ragged Island | 1 | 22.1538 | −75.687 |
| | Adelaine Cay | 1 | 22.1734 | −75.703 |
| | Johnson Cay | 1 | 22.3331 | −75.779 |
| | Nairn Cay | 4 | 22.352 | −75.796 |
| *Puerto Rico* | San Cristobal | 14 | 17.5649 | −67.045 |
| | Rincon | 24 | 18.2101 | −67.159 |
| | Aurora | 3 | 17.9425 | −66.871 |
| | Paraguera | 1 | 17.997 | −67.052 |
| *USVI* | Hawksnest Bay | 6 | 18.3472 | −64.781 |
| | Johnsons Reef | 12 | 18.3617 | −64.774 |
| | Haulover Bay | 13 | 18.3489 | −64.677 |
| | Buck Island | 14 | 18.2774 | −64.894 |
| | Flat Key | 4 | 18.317 | −64.989 |
| | Hans Lollik | 4 | 18.4019 | −64.906 |
| | Sapphire | 6 | 18.3333 | −64.85 |
| | Botany | 3 | 18.3572 | −65.036 |
| *Belize* | unknown | 3 | NA | NA |
| | Bugle Caye | 1 | NA | NA |
| | Curlew | 5 | 16.7909 | −88.083 |
| | Gladden | 1 | 16.4401 | −88.192 |
| | Glovers Atoll | 3 | NA | NA |
| | GSTF1 | 5 | 16.5499 | −88.05 |
| | GSTF12 | 7 | 16.5499 | −88.05 |
| | Larks Caye | 1 | NA | NA |
| | Laughing Bird Caye | 4 | 16.4367 | −88.199 |
| | Loggerhead | 2 | NA | NA |
| | Sandbores | 3 | 16.7791 | −88.118 |
| | Carrie Bow | 13 | 16.8021 | −88.082 |
| *Curacao* | Blue Bay | 7 | 12.1352 | −68.99 |
| | Boka Patrick | 8 | 12.2873 | −69.043 |
| | Directors Bay | 2 | 12.0664 | −68.8603 |
| | East Point | 4 | 12.0407 | −68.783 |
| | PuntuPicu | 9 | 12.0831 | −68.896 |
| | Red Bay | 2 | 12.1355 | −68.99 |
| | Sea Aquarium | 9 | 12.0838 | −68.896 |
| | Water Factory | 3 | 12.1085 | −68.9528 |
| *Sum* | | 260 | | |

**Notes.**
NA, not available.

| | Region | Pool | Coral colonies | Lane | Total reads | Retained Reads after processing | Average number of retained sequence reads per sample | Standard deviation |
|---|---|---|---|---|---|---|---|---|
| | | B1 | 8 | 2 | 50,900,230 | 41,199,646 | 5,149,956 | 1,915,875 |
| | Bahamas | B2 | 8 | 2 | 56,097,984 | 45,237,633 | 5,654,704 | 1,853,265 |
| West | | B3 | 8 | 2 | 58,379,852 | 47,706,860 | 5,963,358 | 2,734,261 |
| | | F1 | 8 | 1 | 50,925,548 | 39,750,070 | 4,968,759 | 1,681,820 |
| | Florida | F2 | 8 | 1 | 48,752,776 | 42,036,153 | 5,254,519 | 4,422,737 |
| | | F3 | 8 | 1 | 49,942,322 | 38,611,895 | 4,826,487 | 2,518,097 |
| | | P1 | 8 | 1 | 43,979,338 | 36,237,997 | 4,529,750 | 4,166,551 |
| | Puerto Rico | P2 | 8 | 1 | 55,267,402 | 47,235,081 | 5,904,385 | 4,096,287 |
| East | | P3 | 8 | 1 | 47,324,190 | 34,835,445 | 4,354,431 | 3,117,707 |
| | | U1 | 8 | 2 | 40,616,766 | 33,170,324 | 4,146,291 | 2,187,597 |
| | USVI | U2 | 8 | 2 | 43,215,386 | 34,291,498 | 4,286,437 | 1,187,166 |
| | | U3 | 8 | 2 | 45,849,098 | 38,439,719 | 4,804,965 | 1,555,938 |
| | | Sum | 96 | | 591,250,892 | 478,752,321 | | |

**Table 2  RAD-tag sequencing summary table of *Acropora palmata* samples.**

sliding-window), adapter contamination, and uncalled bases. Since all indices differed by at least 2 bp, it was possible to correct and retain any index that differed by a single bp from an expected index.

## Assembly

Processed sequences were then aligned to the *Acropora digitifera* genome (V1.0) (*Shinzato et al., 2011*) with Bowtie2 (*Langmead & Salzberg, 2012*) within the Galaxy (*Bedoya-Reina et al., 2013*; *Blankenberg et al., 2014*) framework using end-end read alignment settings in order to remove symbiont and other associated microorganisms. After alignment, paired-end sequencing BAM files were assembled in the ref_map.pl pipeline in STACKS 1.30 with the following parameters. Each paired-end sequencing set was run separately through STACKS to compare results (designated Read1 and Read2) in a one-way ANOVA. The ANOVA used each paired-end read as a technical replicate of the same genomic region. We did this to assess whether we would retrieve similar estimates of $F_{IS}$ and heterozygosity from both reads, as expected.

The number of raw reads required to report a stack was $m = 5$. The number of mismatches allowed between loci when building the catalog was $n = 4$. SNPs with a log-likelihood of less than $-10$ were removed as reads with poor log-likelihoods tend to have sequencing error and/or low coverage. Two of the barcodes (TCGAT and CGATC) had few sequence reads across all four geographic regions with all Illumina lanes being affected, and samples with these barcodes were removed before assembly in Stacks.

Sequencing reads are available under NCBI BioProject ID PRJNA407327.

## Genome coverage

Bedtools (*Quinlan & Hall, 2010*) was used to create a histogram of genome coverage for each sample from the Bowtie2 BAM format alignment files. All positions with a depth

of coverage greater to or equal to 20 were combined into a single bin in the histogram. Data from all geographic regions were averaged (excluding samples with barcodes TCGAT and CGATC) and a cumulative distribution of sequencing coverage was then plotted in SigmaPlot v12.

## Population genetic statistics

We explored values for several parameters relevant to population genetic analyses. In the Populations module in Stacks 1.30 we required a locus to be present in all regions for all analyses (option— $p = 4$). For each locus, we then set the minimum percentage of individuals in a region required to have data for that locus to 40% or 60% (option—r). Further, we set the minimum minor allele frequency (MAF) required to process a nucleotide site at a locus (option—min_maf) to 0.025, 0.05 and 0.075. A $p$-value correction was applied to $F_{ST}$ scores, so that if a $F_{ST}$ score was not significantly different from 0 (according to Fisher's Exact Test) the value was set to 0. Additionally, only one random SNP from any RAD locus was written to the STRUCTURE export file in order to prevent linked loci from being processed. Read 1 and Read 2 STRUCTURE export files were combined and duplicate loci removed randomly between reads. $F_{ST}$ ($p$-value < 0.05) was calculated in STACKS. $F_{IS}$ and $F_{ST}$ distributions are included in the Figs. S1 and S2.

## Clustering analyses

Clustering analyses for the SNP and microsatellite analysis were performed in the program STRUCTURE 2.3.4 (*Falush, Stephens & Pritchard, 2003*; *Hubisz et al., 2009*) using the admixture model with correlated allele frequencies. The analysis included the following parameters: 100,000 burn-in iterations and 1,000,000 Markov chain Monte Carlo repetitions, with and without a population prior, for a total of three replicates for each value of $K$. $K$ values ranged from 2 to 5. The most likely value for $K$ was determined by CLUMPAK (*Kopelman et al., 2015*) BEST K which uses either the Evanno method (*Evanno, Regnaut & Goudet, 2005*) or LN(PR(X|K) values to identify the $K$ for which $Pr(K = k)$ is the highest as described in STRUCTURE's manual section 5.1. Results of the three structure runs were merged with CLUMPAK (*Kopelman et al., 2015*). Based on our exploration of minor allele frequency (MAF) cut off values and the percent of individuals per geographic region allowed to miss a locus (%M), we report results for MAF = 0.05 and for a %M = 60% in the main text (Fig. 2). STRUCTURE clustering analyses for minor allele frequencies cutoffs of 0.025 and 0.075 are included in the Fig. S3. STRUCTURE clustering analysis when the minor allele frequency cutoff was 0.05 and when outlier loci were removed, is also included in Fig. S4. STRUCTURE clustering analysis when the minor allele frequency cutoff was 0.05 and when a locus must be present in at least 40% of individuals in a geographic region, is included in the Fig. S5. PCA clustering analysis, for SNPs and microsatellites, using adegenet (*Jombart, 2008*) is included in the Fig. S6.

Previously genotyped samples ($n = 260$) at 10 and 11 microsatellite markers (181, 182, 192, 207, 0,585, 0513, 2,637, 007, 9,253, 5,047, with and without locus 166) (*Baums et al., 2009*; *Baums, Hughes & Hellberg, 2005a*) were also analyzed with STRUCTURE 2.3.4 (*Falush, Stephens & Pritchard, 2003*; *Hubisz et al., 2009*) using the admixture model with
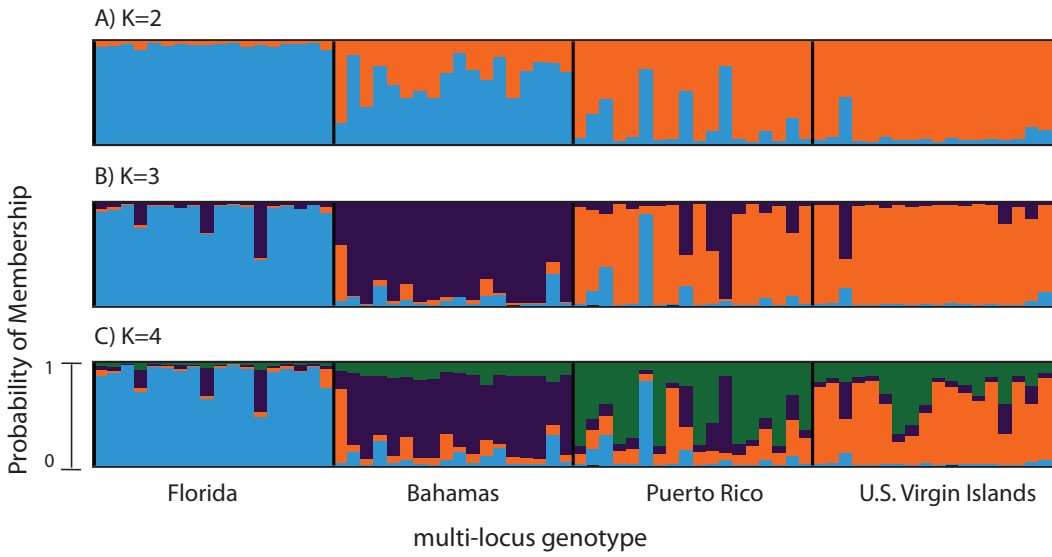

**Figure 2** **Bayesian cluster analysis of 307 SNP loci from *Acropora palmata* ($n = 96$).** Reefs within geographic regions 1–4 sorted by latitude: Florida, Bahamas, Puerto Rico, US Virgin Islands. Analysis included only one SNP per locus after combining Read 1 and Read 2. Shown is the probability of membership ($y$-axis) in a given cluster for each sample ($x$-axis) assuming values of $K = 2$ (A), $K = 3$ (B), and $K = 4$ (C). The most probable $K$ was 3 (B) for the minor allele frequency corrected SNPs based on the mean estimated log probability of the data at a given $K$ (3 replicate runs per $K$, $\pm1$ standard deviation).

correlated allele frequencies (See Table 1 for sample information). The analysis included the following parameters; 100,000 burn-in iterations and 1,000,000 MCMC repetitions, with and without a population prior, for a total of 3 replicates for each value of $K$. $K$ values tested ranged from 2 to 7.

## Mantel tests

Data on temperature, salinity, dissolved oxygen (ml/l), and phosphates was downloaded from the World Ocean Atlas 2013 (WOA13 V2, Table S3). Silicates and nitrates were not used as there was not sufficient data for all locations. For the Bahamas, Puerto Rico, and the USVI the geographic center point among several sampling sites was used because reefs were further apart than in Florida. For all data, the statistical mean of the annual average of years 1955–2012 and depths of 0–10 m was used. Grid sizes were 1/4° for temperature and salinity, and 1° for dissolved oxygen (ml/l), and phosphates (μmol/l) (Table S2). SPSS V22 was used to calculate a dissimilarity matrix expressed as the Euclidean distances between geographic regions based on the above environmental data. To obtain a single GPS location for each geographic region we had to average the latitude and longitude coordinates for all samples in each region. Then GenAlEx v6.501 (*Peakall & Smouse, 2006*) was used to calculate a pairwise geographic distance matrix between the four geographic regions. GenAlEx v6.501 (*Peakall & Smouse, 2006*) was used to calculate Mantel multi-comparison tests between the geographic distance matrix, $F_{ST}$ pairwise matrix between geographic regions from Stacks, and the environmental dissimilarity matrix.

## Outlier analysis

Two independent methods were applied to identify putative loci under selection. The first program used was LOSITAN (*Antao et al., 2008*) which utilizes the method of *Beaumont & Nichols (1996a)* to identify loci under selection based on the joint distributions of expected heterozygosity and $F_{ST}$ under an island model of migration. The following settings were used for the SNP and the microsatellite datasets. The neutral mean setting was selected in which during an initial run (100,000 simulations), a candidate subset of selected loci (outside the 95% confidence interval) were identified and removed. Then the distribution of neutral $F_{ST}$ was computed using 100,000 simulations and a bisection approximation algorithm (*Antao et al., 2008*), with the following options, force mean $F_{ST}$, infinite alleles mutation model, and a confidence interval 0.99. A FDR <0.1 correction for multiple testing was applied. Loci outside the upper and lower confidence areas were identified as candidates affected by positive and balancing selection, respectively (Table S4). All geographic regions were analyzed together. Outliers identified as being under balancing selection were not considered as these are more likely to be false positives (*Lotterhos & Whitlock, 2014*). The positive outlier loci ($p < 0.01$) were blasted against the NCBI nr, UniProt, and Trembl databases with parameters of expected value = 0.00001, gap opening penalty = 11, gap extension penalty = 1, length of initial exact match (word size) = 6 and scoring matrix = BLOSUM62 using BLASTX 2.2.32 + (*Altschul et al., 1997*).

The STACKS exported GENEPOP dataset was also reformatted with PGDSPIDER version 2.0.5.2 (*Lischer & Excoffier, 2012*) to a GESTE file. The method of *Foll & Gaggiotti (2008)* was performed using BAYESCAN 2.0 (http://www-leca.ujf-grenoble.fr/logiciels.html). For each locus, the probability of it being under selection was inferred using the Bayes factor (BF). Based on *Jeffreys*' (*1961*) scale of evidence, a log10 BF of 1.5–2.0 is interpreted as "strong evidence" for departure from neutrality at that locus and corresponds to a posterior probability between 0.97–0.99. For our analysis, the estimation of model parameters was set as 20 pilot runs of 5,000 iterations each, followed by 50,000 iterations.

# RESULTS

## Summary statistics

Illumina sequencing of the RAD libraries generated 49.3 million reads per pool of eight samples, averaging 6.2 million 150 bp reads per sample prior to quality filtering. After quality filtering, 4.99 million reads per sample (81%) were retained on average (Table 2). Pools had similar numbers of reads after processing (mean = 39.9 million per pool, SD = 4.95 million, one-way ANOVA, $F = 2.638$, $p > 0.1$). The average % GC content for Read 1 and 2 was 41.7 and 39.6, respectively. The percentage of polymorphic sites per genomic region varied little among geographic regions, from 0.150 to 0.173% (Table 3). The average observed heterozygosity in variant sites was 22%. Overall $F_{IS}$ values, when considering all sites with a minor allele frequency cutoff a $\geq 0.05$, were close to 0 and hence provided no evidence of inbreeding (Table 3). However, when only considering variant positions within the region of Florida, $F_{IS}$ values were negative ($F_{is} = -0.0086$), indicating an excess of heterozygosity. Using the two paired-end read sets as replicates, a one-way ANOVA was

**Table 3  Summary statistics for Read 1 and Read 2 combined.** $F_{IS}$ calculations with and without minor llele frequency restrictions. Calculated by STACKS 1.30.

| | | Bahamas | Florida | Puerto Rico | USVI |
|---|---|---|---|---|---|
| All positions: variant and fixed | Total Sites | 200425 | 200425 | 200425 | 200425 |
| | Variant Sites | 390 | 390 | 390 | 390 |
| | Private Alleles | 2 | 1 | 0 | 2 |
| | % PL | 0.1732 | 0.1497 | 0.1694 | 0.1668 |
| | $F_{IS}$ | 0.00005 | 0 | 0 | 0.00005 |
| | Nucleotide diversity ($\pi$) | 0.0004 | 0.0004 | 0.0004 | 0.0004 |
| Variant positions only | Obs Hom | 0.7728 | 0.7874 | 0.7791 | 0.7815 |
| | Std Err | 0.0164 | 0.0164 | 0.0154 | 0.0154 |
| | Obs Het | 0.2273 | 0.2126 | 0.2210 | 0.2186 |
| | Std Err | 0.0164 | 0.0164 | 0.0154 | 0.0154 |
| | Exp Hom | 0.7832 | 0.8050 | 0.7919 | 0.7916 |
| | Exp Het | 0.2169 | 0.1951 | 0.2081 | 0.2085 |
| | $F_{IS}$ | 0.02235 | **−0.0086** | 0.0035 | 0.02065 |
| | Nucleotide diversity ($\pi$) | 0.2254 | 0.2034 | 0.2174 | 0.21705 |

**Notes.**

% PL, percent polymorphic loci; Obs Hom, observed homozygosity; Obs Het, observed heterozygosity; StdErr, standard error; Exp, expected.

Bold values are significantly different from 0 ($P < 0.05$).

**Table 4  Pairwise $F_{ST}$ comparisons of geographic regions based on SNP (A) and microsatellite (B) data.**

| | Bahamas | Florida | Puerto Rico | USVI |
|---|---|---|---|---|
| (A) | | | | |
| Bahamas | | | | |
| Florida | 0.018 | | | |
| Puerto Rico | 0.013 | 0.022 | | |
| USVI | 0.018 | 0.022 | 0.009 | |

| | Belize | Florida | Bahamas | Puerto Rico | USVI | Curacao |
|---|---|---|---|---|---|---|
| (B) | | | | | | |
| Belize | | | | | | |
| Florida | 0.0040 | | | | | |
| Bahamas | 0.0115 | 0.0097 | | | | |
| Puerto Rico | 0.0206 | 0.0153 | 0.0063 | | | |
| USVI | 0.0206 | 0.0174 | 0.0098 | 0.0037 | | |
| Curacao | 0.0240 | 0.0138 | 0.0181 | 0.0173 | 0.0208 | |

performed for each variable (Table 4). Summary statistics for all geographic regions were found to be similar. Alignment of *A. palmata* SNPs to the published *A. digitifera* genome indicated that on average, 2.5% percent of the *A. digitifera* genome had sequence coverage at a stack depth of 5 (Fig. S7). All four geographic regions produced similar sequence coverage.

## Population genetics

A total of 390 SNPS were identified after filtering and including a minor allele frequency cutoff $\geq 0.05$ (Table 3). This included 219 for Read 1 and 176 for Read 2 from the paired-end sequencing (5 SNPs were identical between reads and were only considered once). Analysis of Molecular Variance (AMOVA) revealed patterns of significant genetic differentiation among geographic regions (Table 4). This was also evident when the 307 SNPs (analysis included only one SNP per 150 bp locus) were subjected to a multi-locus clustering analysis in STRUCTURE. Samples from Florida clustered first, followed by the Bahamas at $K = 3$. Puerto Rico and the USVI were not distinguishable until $K = 4$, (Fig. 2). Clumpak Best K (Kopelman et al., 2015) indicated that $K = 3$ was the most likely $K$-value, after both the Evanno method and LN(PR(X|K) values, regardless of whether the geographic region was used as a prior.

To compare to the SNP analysis, microsatellite data from samples collected in six regions were analyzed in STRUCTURE using the geographic region as a prior (Table S5). At $K = 2$, a western (including Belize, Florida, Bahamas and Puerto Rico) and an eastern cluster (including the USVI and Curacao) was evident (Fig. 3A). At $K = 3$, an isolation-by-distance like pattern was apparent in the western cluster (Fig. 3B). $K = 4$ was the most likely $K$-value, after both the Evanno method and LN(PR(X|K) values, based on 11 microsatellite markers (Kopelman et al., 2015). Florida and Belize grouped as one cluster, and Puerto Rico and the Bahamas as the second, with the USVI as the third and Curacao as an admixed fourth cluster (Fig. 3C).

According to the outlier analysis in LOSITAN, microsatellite locus 166 was identified as a potential outlier and thus possibly under selection. It was therefore excluded from the analysis in STRUCTURE. This resulted in more comparable results to the SNP analysis with the most likely $K$-value being 3, after both the Evanno method and LN(PR(X|K)) values (Kopelman et al., 2015). Again, the first separation was between a western and an eastern cluster, however this time Puerto Rico assigned to the eastern cluster with an isolation-by-distance like pattern appearing between the west and east (Fig. 3D). At the most likely $K$ of 3, Curacao now formed a separate cluster. At $K = 4$, the Bahamas started to separate from the remainder of the western region similar to what was observed in the SNP clustering analysis (Fig. 3E).

## Environmental drivers of population structure

A Mantel test showed a significant positive relationship in the SNP dataset between pairwise $F_{ST}$ values and geographic distance ($R^2 = 0.65$, $p = 0.05$) consistent with the microsatellite results (10 loci) from the Florida, Bahamas, Puerto Rico, and Curacao samples only (Figs. 4C and 4D). Correlations between environmental factors including average temperature, salinity, dissolved oxygen, and pairwise $F_{ST}$ values or geographic distance were not significant (Figs. 4A and 4B).

## Loci under selection

BayeScan and LOSITAN identified 2 and 12 SNPs (Table S4) that showed signs of positive selection when including all four geographic regions, one of which was identified by

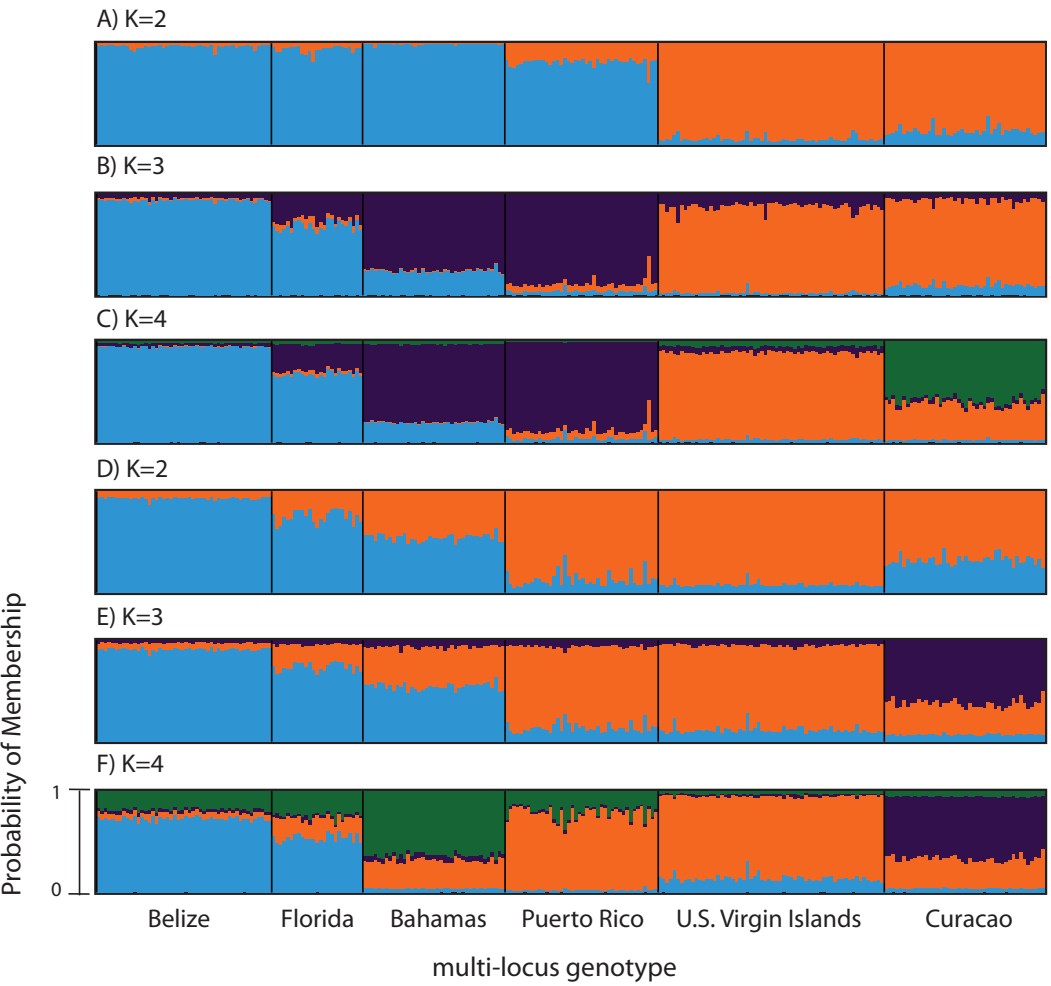

**Figure 3** Bayesian cluster analysis of microsatellite data from *Acropora palmata* (*n* = 260). (A–C) Analysis of 11 microsatellite loci with the most probable *K* being 4. (D–E) Exclusion of the outlier locus 166 resulted in the analysis of 10 microsatellites with the most probable *K* being 3. Shown is the probability of membership (*y*-axis) in a given cluster for each sample (*x*-axis) assuming values of *K* = 2 (A, D), *K* = 3 (B, E), and *K* = 4 (C, F).

both programs (a total of 13 unique loci identified between both programs). Outliers accounted for 3.3% of the total SNPs, consistent with other studies in which $F_{ST}$ outlier loci have represented a substantial fraction of the total loci investigated (2–10%) (*Nosil, Funk & Ortiz-Barrientos, 2009*). Annotation of the candidate loci proved difficult as only 23% produced significant hits when queried against the NCBI NR database, Uniprot, and Trembl; with two of the hits being annotated as unconventional myosin-IXb isoform X7 and tyrosine-protein kinase transmembrane receptor ROR1-like. Screening of the microsatellite loci identified locus 166 as an outlier under positive selection, yet no annotation information of this locus is currently available.

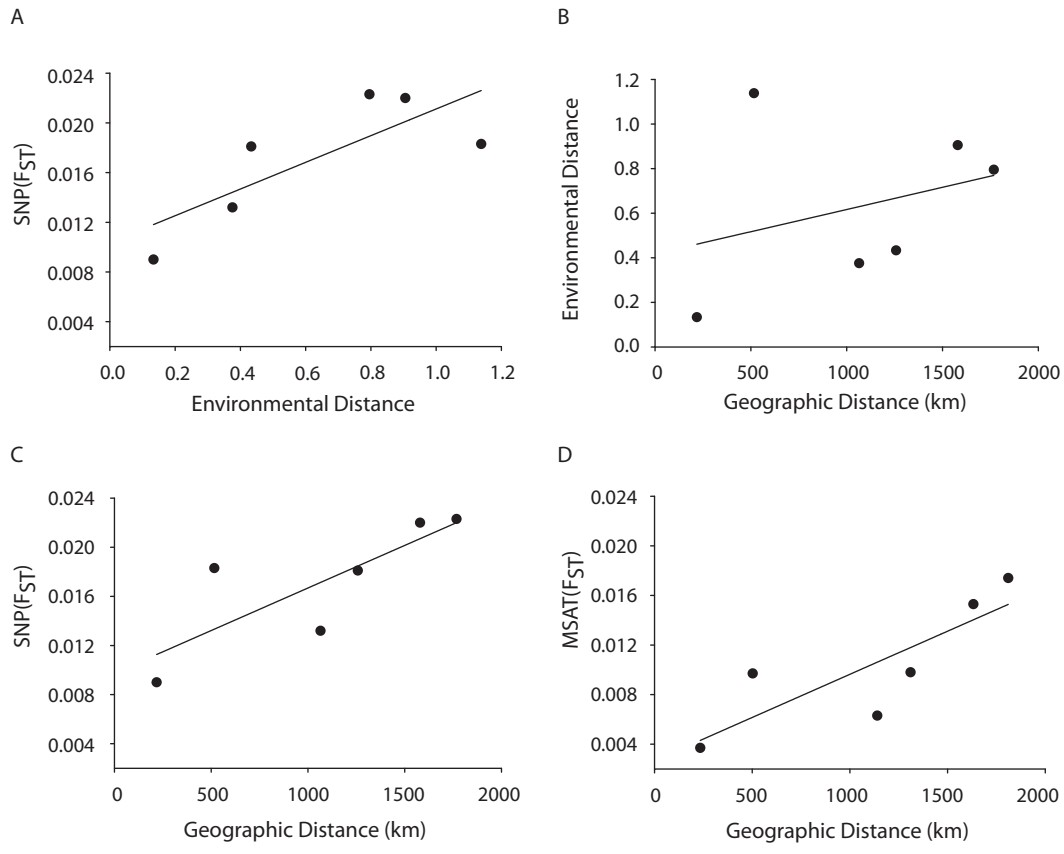

**Figure 4** **MANTEL matrix correlation test between genetic ($F_{st}$), environmental (Euclidean) and geographic distances (km).** *Acropora palmata* samples from four geographic regions (Florida, Bahamas, Puerto Rico and USVI) were genotyped with 307 SNP (A–C) or 10 neutral microsatellite markers (D). (A) $y = 0.0107x + 0.0104$, $R^2 = 0.610$, $p = 0.09$. (B) $y = 0.002x + 0.4175$, $R^2 = 0.101$, $p = 0.21$. (C) $y = 0.000007x + 0.0098$. $R^2 = 0.648$, $p$-value=0.05. (D) $y = 0.000007x + 0.0027$. $R^2 = 0.69$, $p = 0.04$.

## DISCUSSION

### Comparison with previous *Acropora* gene flow studies

The previous range-wide survey of *A. palmata* population genetic structure using five, presumed neutrally evolving microsatellite markers showed that while most reefs are self-recruiting, *A. palmata* stands are not inbred and harbor high microsatellite genetic diversity (*Baums, Miller & Hellberg, 2005b*). Furthermore, *A. palmata* stands were structured into two long-separated populations, one in the eastern and one in the western Caribbean (*Baums, Miller & Hellberg, 2005b*). Here, we report that genome-wide SNPs (MAF ≥ 0.05) resolved further population structure in the endangered reef-building coral, *A. palmata* from Florida to the USVI compared to previous microsatellite-based analyses.

It was recently suggested that the East-West divide of *A. palmata* lies not in the Mona Passage (*Baums, Miller & Hellberg, 2005b*; *Baums, Paris & Cherubin, 2006b*) but rather to the east of Puerto Rico (Fig. 1, *Mège et al., 2014*). The 307 SNPs analyzed here confirm earlier findings that Puerto Rico and the USVI regions are more similar to each other

than Puerto Rico is to either the Bahamas or Florida without imposing any priors in a STRUCTURE analysis (MAF $\geq$ 0.05). However, it is not always possible to determine, with confidence, the correct clustering solution that accurately reflects genetic population structure when there is an underlying isolation by distance pattern (*Frantz et al., 2009*). We show here that there is significant isolation by geographic distance from Florida to the USVI when using presumably neutrally evolving SNP and microsatellite loci. Interestingly, inclusion of microsatellite locus 166, flagged as being an outlier locus, obscured this isolation by distance pattern (Fig. S8). Therefore, locus 166 is a strong candidate for a locus under selection (or it is linked to a locus under selection) and its functional significance might prove a fruitful subject for future studies (*Nielsen, Hansen & Meldrup, 2006*).

An east–west Caribbean divide was also evident in the corals *Orbicella annularis* (*Foster et al., 2012*) and *Acropora cervicornis* (*Vollmer & Palumbi, 2007*). An additional barrier to gene flow in *A. palmata* was reported by *Porto-Hannes et al. (2015)* between Venezuela and the Mesoamerican Barrier Reef System utilizing four of the microsatellites markers.

The total number of SNPs ($n = 307$) retained for population genetic analysis was lower than expected. This was due to more than a 10-fold increase in the number of fragments retrieved from the genome digest using the enzymes MluCI (´AATT) and NlaIII (CATG´) compared to what was predicted from an in-silico restriction of an incomplete draft genome of *A. palmata* (Baums, unpublished). The in-silico restriction predicted 19,067 to the actual 322,425 (read 1) and 276,753 (read 2) fragments retrieved. This under-prediction was most likely due to an early, incomplete genome draft and unknown genome size at the time of this study (*Herrera, Reyes-Herrera & Shank, 2015*). A larger set of SNP loci may reveal additional finer scale structure in *A. palmata* across the Caribbean. However, this may not necessarily be the case. In a study that used three orders of magnitude more loci (905,561 SNPs) failed to reveal population structure in *A. digitifera* collected from the Ryukyu Archipelago of Japan using Bayesian clustering based methods (*Shinzato et al., 2015*). Low coverage, 5× in this study, is also a concern however this depth of coverage has been used in other non-model species (*Babbucci et al., 2016*; *Blanco-Bercial & Bucklin, 2016*; *Laporte et al., 2016*). Yet in the coral *Platygyra daedalea,* 5× coverage was sufficient to assign samples to two distinct clusters based on their geographic origin, the Persian Gulf or Sea of Oman and was consistent with their 20× coverage data set (*Howells et al., 2016*).

A Mantel test showed a significant positive relationship between the SNP-derived pairwise $F_{ST}$ values and geographic distance ($r^2 = 0.65$, $p = 0.05$) consistent with the microsatellite results (10 loci) from the Florida, Bahamas, Puerto Rico, and Curacao samples (Figs. 4C and 4D). This may be due to Wright's Isolation-By-Distance (IBD) process however Mantel tests are prone to false positives as the test assumes spatial independence of the data (*Meirmans, 2012*). Nevertheless, genetic variability is structured in geographic space.

Correlations between environmental factors including average temperature, salinity, dissolved oxygen, and pairwise $F_{ST}$ values or geographic distance were not significant (Figs. 4A, 4B). It should be noted that the environmental data had a resolution of $\frac{1}{4}$ to 1 degree latitude, an equivalent of about 28–111 km, whereas the genetic data was collected on much smaller spatial scales. For example, in Florida, sampled reefs were often less

than 10 km apart, and the distance between Sand Island Reef and French Reef is only 2.6 km. (Table S2). Here, reefs often harbor just one or a few *A. palmata* genets (albeit represented by many colonies) making it challenging to obtain the needed >25 genets per population recommended for $F_{ST}$ analyses on a scale of a few km. Thus, genets were pooled over geographic regions to match the scale of the environmental data and yield sample sizes of at least 25 per location. Yet significant micro-environmental differences among colonies growing on the same reef have been documented (*Drury, Manzello & Lirman, 2017*; *Gorospe & Karl, 2010*). Therefore, landscape genetic approaches that may reveal environmental drivers of population differentiation (*Manel et al., 2003*) must await higher resolution environmental data and, perhaps, a greater number of SNP loci.

## Genetic diversity indices in *A. palmata*

Several factors could account for negative $F_{IS}$ values including negative assortative mating, if a species is outcrossed and lacks selfed progeny or there is a selection pressure that favors the most heterozygous genets. Of our samples, 49 out of 96 were ramets of larger genets. *A. palmata* colonies fragment frequently; the branches regrow into new colonies resulting in stands of genetically identical colonies (*Baums, Miller & Hellberg, 2006a*). (Note that samples included here all represented distinct genets). Asexual reproduction could explain the excess of heterozygosity in *A. palmata* within the Florida region (see *Balloux, Lehmann & De Meeus, 2003*; *Carlon, 1999*; *Delmotte et al., 2002*). Excess hetereozygosity has been observed in other clonal organisms. For example, significant negative $F_{IS}$ values in a partially clonal but self-incompatible wild cherry tree was explained in part by asexual reproduction (*Stoeckel et al., 2006*).

Nucleotide diversity is a measure of a species' genetic diversity and varies predictably with life history (*Hamrick & Godt, 1996*; *Romiguier et al., 2014*). Because *A. palmata* populations experienced dramatic losses in the 1980s and therefore may now have reduced genetic diversity we compared *A. palmata*'s nucleotide diversity to the diversity found in other species. The nucleotide diversity π, describes the degree of nucleotide polymorphism in populations and can be calculated based on variant sites only or on variant and non-variant sites combined. In acroporids, estimates range from 0.007–0.022 (*Macdonald, Schleyer & Lamb, 2011*) in *A. austere* to 0.09 in *A. cervicornis* (*Drury et al., 2016*). In other Cnidaria, estimates range from 0.00403 in *Aiptasia* (*Bellis, Howe & Denver, 2016*) to 0.0065 in *Nematostella* (*Putnam et al., 2007*). Synonymous nucleotide diversity ranged from 0.012–0.020 in transcriptomes from three gorgonian species (*Romiguier et al., 2014*). Average pairwise nucleotide diversity in other metazoans include *Drosophila pseudoobscura* (0.0024–0.0179, *Kulathinal, Stevison & Noor, 2009*) and *Homo sapiens* (0.000751, *Sachidanandam et al., 2001*). Our estimates of nucleotide diversity (including variant and non-variant sites) was 0.0004 for all geographic regions, an order of magnitude lower than in other cnidarians. Further, based on a survey of 374 individual transcriptome-derived SNPs from 76 non-model animal species, the level of nucleotide diversity found in *A. palmata* is well below that predicted for a long-lived species, with small propagule size and large adult size (*Romiguier et al., 2014*). This low nucleotide diversity could be due to either a relatively small long-term effective population size, a severe bottleneck associated with a selective
sweep (*Ellegren & Galtier, 2016*), the small number of SNPs included in this study (*Fischer et al., 2017*) or the RAD-tag method (*Arnold et al., 2013*). In addition, we find that Florida is the least genetically diverse geographic region when comparing nucleotide diversity in variant sites only (0.203, Table 3), as would be expected in a marginal environment (*Baums, 2008*; *Baums et al., 2014a*; *Cahill & Levinton, 2016*; *Eckert, Samis & Lougheed, 2008*). This is in contrast to *Drury et al. (2016)*, which found samples of the congener *A. cervicornis* from Florida to be higher in SNP nucleotide diversity than those from the Dominican Republic. Increased sampling of the genome as well as analysis of historical samples may shed light on whether the low nucleotide diversity in *A. palmata* is due to technical issues, the recent population bottleneck or unrelated causes.

Allelic richness of microsatellite data correlates better with genome-wide estimates of genetic diversity based on SNPs than heterozygosity (*Fischer et al., 2017*) and allelic richness is more sensitive to recent population bottlenecks than heterozygosity (*Allendorf, 1986*). Average microsatellite-based allelic richness in 14 Indo-Pacific *Acropora* corals was 4.96 overall and 6.21 in the five geographically widespread species (calculated based on Table 6 in *Richards & Oppen, 2012*) which compares favorably with an average allelic richness of 8.49 in *A. palmata* found here. Thus, allelic richness of microsatellite loci remains high in Caribbean *A. palmata* despite recent population declines and the documented loss of alleles in Florida (*Williams, Miller & Baums, 2014*).

To resolve the contradictory findings with respect to genetic diversity based on micorsatellites and SNPs, future studies should include several thousand SNPs assayed in samples from across the species range. This approach may provide more conclusive data on the impact of recent population declines on overall genetic diversity in *A. palmata*.

## Genes under positive selection

Thirteen loci out of 395 were identified as being under positive selection in *Acropora palmata*. Detecting regions of the genome under selection is difficult, and statistical detection methods are prone to different rates of type 1 and type 2 errors. Further, LOSITAN and BayeScan often identify different loci as being under selection (*Narum & Hess, 2011*). LOSITAN identifies outliers based on the joint distributions of $F_{ST}$ and expected heterozygosity under an island model of migration (*Beaumont & Nichols, 1996b*). Whereas, BayeScan uses a hierarchical Bayesian method of *Foll & Gaggiotti (2008)*, which has been modified based on the approach proposed by *Beaumont & Balding (2004)*. *Lotterhos & Whitlock (2014)* claim that many of the published $F_{ST}$ outliers based on FDIST2 and BayeScan are probably false positives; however, their results show that these false positives are mostly in balancing selection and we did not include outliers identified as being under balancing selection for this reason. In a comparison of $F_{ST}$ outlier tests, FDIST2 and BayeScan appeared to provide the most power, depending on the scenario, and BayeScan had fewest false positives (*Narum & Hess, 2011*). Here, one locus 80994_17 (Digitifera scaffold gi|342271542|dbj|BACK01025553.1|, basepair = 5,143) out of 13 was identified by both programs, therefore we consider this locus to be a strong candidate for being under selection and the other loci as possible candidates. However, Stacks locus 80994_17 was not annotated, a common occurrence even for transcribed loci in corals,

where typically a third or less of genes have annotation (*Meyer, Aglyamova & Matz, 2011*; *Polato et al., 2010*).

One of the SNP loci identified as being under positive selection was annotated as a *tyrosine-protein kinase transmembrane receptor ROR1-like.* ROR receptor protein is associated with the nervous system in the fruit fly *Drosophila* (*Wilson, Goberdhan & Steller, 1993*), nematode *C. elegans* (*Francis et al., 2005*), *and sea slug Aplysia californica* (*McKay et al., 2001*). Functional analysis of *cam-1*, a gene that encodes for a ROR kinase in *C. elegans*, demonstrated roles in both the orientation of polarity in asymmetric cell division and axon outgrowth, and the ability to guide migrating cells (*Forrester et al., 1999*). The role of *ROR1* receptors in Cnidaria is unknown although studies in *Hydra* suggest a function in regulating cell specification and tissue morphogenesis (*Bertrand, Iwema & Escriva, 2014*; *Krishnapati & Ghaskadbi, 2014*; *Lange et al., 2014*).

Another SNP identified as being under positive selection was located in the gene annotated as unconventional *myosin-IXb isoform X7*, a Rho GTPase-activating protein (RhoGAP) that is essential for coordinating the activity of Rho GTPases. Invertebrates are thought to contain a single myosin class IX gene (the exception is *Drosphilia* which has none) whereas most vertebrates have two with fishes having four (*Liao, Elfrink & Bähler, 2010*). In general, Rho GTPases control the assembly and organization of the actin cytoskeleton which includes many functions such as cell adhesion, contraction and spreading, migration, morphogenesis, and phagocytosis. Little is known about the function of myosin-IX in invertebrates. However, a recent study in which *Orbicella faveolata* were exposed to immune challenges identified Unconventional myosin-IXb as a transcript that was significantly correlated with melanin protein activity (*Fuess, Weil & Mydlarz, 2016*). In humans, Myosin-IXb is highly expressed in tissues of the immune system such as the lymph nodes, thymus, and spleen and also in immune cells like dendritic cells, macrophages and CD4 + T cells (*Wirth et al., 1996*). Myosin-IXb knockout mice showed impaired recruitment of monocytes and macrophages when exposed to a chemoattractant demonstrating that Myosin-IXb has an important function in innate immune responses *in vivo* (*Hanley et al., 2010*). Because statistical screens for loci under selection carry a high rate of false positive results, further experimental evidence is necessary before these loci can be considered targets of selection.

## Restoration implications

Restoration efforts should proceed under the assumption that *A. palmata* harbors a significant amount of population structure requiring close matches of collection and outplant sites. Hybridization of *A. palmata* from different geographic regions may or may not result in heterosis depending on sexual compatibility, but would be worth pursuing in an *ex situ* setting to enable close monitoring of offspring performance under elevated temperatures (*Van Oppen et al., 2015*). With respect to the sharply declining Florida colonies, these findings underline the need to manage and restore Florida's *A. palmata* as an isolated, genotypically depleted geographic region (*Williams, Miller & Baums, 2014*).

## ACKNOWLEDGEMENTS

Thanks to PSU genome sequencing facility for expert library preparation and sequencing. N Polato contributed to the study design. We gratefully acknowledge the efforts by our collaborators around the Caribbean that have contributed samples over the years. We thank the editor and three reviewers for insightful comments.

### Funding

Funding was provided by the NOAA Coral Reef Conservation Program Grant number NA13NOS4820029 and National Science Foundation grants OCE-1537959 and OCE-1516763 to Iliana B. Baums. The funders had no role in study design, data collection and analysis, decision to publish, or preparation of the manuscript.

### Grant Disclosures

The following grant information was disclosed by the authors:
NOAA Coral Reef Conservation Program: NA13NOS4820029.
National Science Foundation grants: OCE-1537959, OCE-1516763.

### Competing Interests

The authors declare there are no competing interests.

### Author Contributions

- Meghann K. Devlin-Durante conceived and designed the experiments, performed the experiments, analyzed the data, wrote the paper, prepared figures and/or tables, reviewed drafts of the paper.
- Iliana B. Baums conceived and designed the experiments, contributed reagents/materials/analysis tools, wrote the paper, prepared figures and/or tables, reviewed drafts of the paper.

### DNA Deposition

The following information was supplied regarding the deposition of DNA sequences:
The Illumina sequences are available under NCBI Bioproject PRJNA407327.

### Supplemental Information

Supplemental information for this article can be found online at http://dx.doi.org/10.7717/peerj.4077#supplemental-information.

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
