# Peer review of "Genome-wide survey of single-nucleotide polymorphisms reveals fine-scale population structure and signs of selection in the threatened Caribbean elkhorn coral, Acropora palmata"

_PeerJ, doi:10.7717/peerj.4077_

## Round 0.1 · original submission · Major Revisions

In this manuscript, the authors develop new single nucleotide polymorphism (SNP) loci for the coral Acropora palmata. These loci are then used to infer population structure for this species in the Caribbean, and test competing hypotheses about the position of an east-west population split based on previously-published microsatellite studies of this species. The three external reviewers all considered this a valuable study with important implications for studying the population structure of Acropora palmata, and broader implications for coral management. The three reviewers did, however have concerns about the clarity, data quality, data selection, and analyses in portions of the manuscript that will require significant revision. In particular, the authors need to address the 5X genome coverage, which one reviewer described as "alarmingly low". The manuscript does mention one paper in which 5X coverage yields results consistent with a 20X coverage, but this topic deserves a more in-depth analysis of potential effects, and perhaps analyses of regions with deeper coverage. Similarly, there were questions about how parameters such as geographic distance were determined, and how analytical choices, such as use of loci present in at least 60% of individuals, and minor allele frequencies of 0.05 were chosen, and potential effects of different reasonable choices. One reviewer also suggested that a map could be useful for clarifying competing hypotheses, and illustrating results, geographic relationships, and distances etc. Finally, please address the limitation of the analysis of SNPs potentially under selection. Please be sure to address each of the reviewers concerns in the cover letter submitted with your revised manuscript. I have attached a PDF with some minor comments and editorial suggestions

Reviewer 1 ·

Basic reporting

Durante and Baums report on the genetic diversity and population genetic relationship of the coral Acropora palmata from a portion of this species' distribution in the Caribbean. The overall presentation is clear and the tables and figures are of good quality. The authors compare results from a small panel of microsatellite loci with genome-wide SNPs derived from genotype-by-sequencing. The authors suggest these results are important for considerations regarding coral connectivity and management. Overall, I found the description of the methods, thoroughness of the analysis and interpretation of the results to be quite limited. The shortcomings of microsatellite analysis, particularly of small number of loci, are now generally known and likelihood of detecting selection with confidence (as opposed to genetic drift or demographic impacts) is even more limited with a relatively low sample size from each location. The RAD-tag data could be utilized more effectively to address some of the authors hypotheses, particularly about relationships of individuals and populations with respect to geographic hypotheses related to the Mona Passage.

Experimental design

The experimental design is adequate by genotyping individual corals from a number of locations. I detail major concerns below as well as indicate positions in the manuscript where additional clarification would be useful.

1. RAD-tag methodology. The authors should provide additional details and support for their approach to RAD-tag. It is unclear from the current text how the libraries were pooled and sequenced (Lines 132-135). In Lines 163-166 it would be helpful for the authors to provide references and support for these filtering steps. It is unclear how and why the authors came to the decision to only use loci present in at least 60% of individuals and a minor allele frequency of 0.05. Similarly, I also found it surprising the authors only used STRUCTURE based approaches when addressing individual data. Why did the authors not use phylogenetic based methods to look at relationships of individuals across the sites? The 5X coverage is alarmingly low (which the authors acknowledge on Line 335) for interpreting their results and I would suggest the authors compare results using a data with lower and greater coverage (based on Figure 1).

2. Mantel tests and environmental data. The authors correlate the FST values with geographic distance and a combination of environmental data. The authors should provide a methodology for how the geographic distance was determined and support for their approach. The environmental data comparison is an interesting comparison but the value of the confidence in these correlations appears suspect. The authors note (Line 280-282) that these environmental data are coarse, which suggests to this reviewer they could be misleading when compared with the geographic distance as a potential explanation for the genetic differences between populations.

Validity of the findings

There are a few places in this manuscript where the findings could be described more clearly or in a more balanced way.

1. Outlier analyses and results. The identification of one microsatellite locus as under potential selection was interesting but given the low number of loci investigated it is difficult to interpret the confidence in this determination. The authors used standard approaches for detecting potential SNPs under selection but the results from these are equivocal. The different methods produced largely unique sets of potential SNPs and the authors should provide an explanation for this difference. Detecting regions of the genome under selection is difficult and I think these results would suggest the possibility that they are false positives. When combined with the data that these loci do not strongly match to any transcribed parts of the genome I would suggest these data and the corresponding Discussion section were removed from the manuscript.

2. Genetic diversity comparisons. The authors invest a significant portion of the Discussion on comparisons of genetic diversity between animal species. These comparisons should be prefaced with a number of caveats about where these numbers were derived (i.e., lab strain, population-averages, etc.). More importantly, in Lines 372-375, the authors correctly state that their estimate from this species could simply be due to the small sample size. This would be a more parsimonious explanation. I think putting in this explanation (a technical one based on methodology) at the end of a more complex biological explanation is not a balanced discussion.

Additional comments

Small comments and suggestions

Table 3. Fis should have capitalized "is"

Figure 2 and 3. I would suggest the authors have a STRUCTURE analysis of with and without SNPs potentially under selection, similar to the microsatellites. Also, the authors should include a STUCTURE analysis of the microsatellite data when comparing only the same sites and individuals as used for SNP analysis.

Figure 4. Please check the figure legend for this figure. I assume panel (b) is also microsatellite data but not clear from the legend. Also, please label y-axis on panel (d).

Supplemental Table 3. It is unclear why some of the sequences in this table have "Ns". I am concerned that these positions were called as SNPs although they are apparently not identified base positions.

Reviewer 2 ·

Basic reporting

The two data files in Supplementary Info that involve microsatellite calls "...STRUCTURE.txt" are unreadable, formatting problem.

In general the paper is well-written but there are a few ways in which it could be clarified.

1. the abstract mentions "an extended set of 12 microsatellite markers" but in Introduction authors refer to "...structure derived from ten and eleven microsatellite loci." First of all, address the mismatch between the 11 loci that were examined and the abstract. Second, to put this ambiguity at the end of the introduction was confusing and unnecessary. My understanding from the rest of the paper is that all of these samples were genotyped at 11 loci - and any dataset may have some missing data, after all - and then in the present analysis one of those loci was determined to be an outlier, so the analysis was repeated using only the 10 non-outlier loci. This is also quite typical, so it will make more sense if you say you used 11 loci, and then re-analyzed after the outlier analysis.

2. another confusing way this was written - Methods/Library Preparation you discuss concentrations without complete units, e.g. ng should be ng/µl, and then "Samples were pooled into four libraries..." please make it clearer that all libraries (individual coral colonies) are uniquely barcoded, to distinguish this from pooled-RAD-seq in which Fst can be determined but not individual-level statistics. This was confusing to me at several points in the manuscript.

otherwise the Intro/background writing is good and supports this paper well. There are places in the Results, e.g. lines 280-285, that are probably more appropriate for the Discussion section of the paper.

Table 1 explain the distinction between A (SNPs) and B (microsatellites) in the legend. Also explain the missing latitude/longitude data?

Experimental design

Experimental design and question are straightforward. A large dataset of individuals with 10 microsatellites are analyzed in the context of a RAD-seq study on a subset of these individuals from 4 primary geographic regions. This analysis allows the determination of any outlier (perhaps locally adapted) diversity, and re-assessing hypotheses of spatial variation in Acropora palmata. For some readers who are less familiar with this region/system, a map may be helpful to distinguish between the competing hypotheses. Genotyping, sequencing, and bioinformatic approaches are typical and appropriate.

Validity of the findings

The overall result - concordance of SNP and microsatellite data once outlier loci are removed and considered separately - is a useful contribution to understanding spatial distribution of diversity in Acropora palmata and other species in the region. Clarifying some of the reporting/writing as mentioned earlier will make these results make better sense to the reader; again I think a map contrasting the hypotheses and the results in this paper may be of value.

Make clear where the Supplementary Figure fits into the general story - that is the IBD analysis when all 11 loci are used, not referred to in manuscript. All the IBD tests are relatively weak with only 6 data points (pairwise contrasts) but with the Structure analyses this is a working hypothesis for the pattern of diversity until more data come along. The argument in lines 280-285 should move to Discussion, but I also don't agree that the spatial scales are so distinct - if the environmental data have spatial resolution of 0.25-1° latitude, that is about 28-111km, not so disparate from the spatial separation of reefs within the 4 regions? So please clarify that argument perhaps with more information on those spatial separation distributions.

I would also consider Meirman 2012 when evaluating the approach to IBD, the consideration of outlier loci under an island model, and so on for interpreting these results at this scale.

The second paragraph under "Genetic diversity indices in A. palmata" is confusing. You are mixing diverse summary statistics, and it isn't clear until late in the paragraph why the specific examples are being chosen - there are many 1000s of estimates of nucleotide diversity in metazoans, and even synthetic analysis across multiple cnidarian species. The contrast with the Romiguier paper starts to clear up what your goal was with those paragraphs, ostensibly to tie observed diversity with known life history to suggest that diversity is quite low in this particular species, which may be an effect of recent devastating losses in A. palmata among other things. I would clarify these paragraphs tremendously, recognize that this type of comparison is common especially in the plant literature (e.g. older work by Hamrick and Godt), but that of course every species has idiosyncratic issues driving their diversity. So just make it more apparent up front what this paragraph/section is attempting to communicate.

Additional comments

Overall a nice contribution that will be a useful resource for Caribbean biogeography and coral diversity projects in the future once it is improved and clarified.

Reviewer 3 ·

Basic reporting

The article is mostly written in clear professional english. There are some parts where grammar needs to be corrected to improve clarity:

- Line 22. Suggest substituting the word 'place' for 'infer'.
- Lines 37-39. Should be split into two separate sentences.
- Line 52. 'permitting agencies'? should it be 'managers' instead?
- Lines 62-63. Not clear meaning
- Lines 91-94. Statements are redundant
- Line 163. 'More' than what?
- Line 172. Replace 'within' with 'in'
- Lines 251-253. This sentence is awkward, suggest splitting in two to make more clear.
- Lines 330 - 334. Sentences are contradictory, not sure what the overall message here is.

The literature references are appropriate, however, given the emphasis on the authors place on the novel use of RAD markers in corals, these results should be placed in a broader context of studies that have used this technic in other animals, particularly invertebrates and non-scleractinian anthozoans.

The results are relevant to test the population structuring hypothesis presented (Mona Passage as a location of major genetic discontinuity)

The article is wells structured. Figures and tables are appropriate, however, Figure 1 should be supplementary as it does not contain critical information to deliver the main message of the paper.

Specific comments:

- Tables 1 and 4. Define in legend what A) and B) indicate

-Figure 3. Is not clear what each pane represents. There are references to (a-c), (d-f) and (D-E). Please make sure it is clear how they are grouped

Experimental design

The article constitutes an original contribution to our understanding of the mechanisms that structure the genetic diversity of species in the ocean. It also provides a nice example where a previous result is reevaluated and updated through the use of newer methods that increase resolution and accuracy. As such, this manuscript nicely fills a knowledge gap in marine biology with conservation implications.

The research was conducted to an acceptable technical (see comments below) and high ethical standard. The methods are described in detail, however it would be useful to see the actual commands run as supplementary material to ensure its reproducibility.

Specific comments:

- Are the samples utilized for RAD-seq the same as the ones used for microsatellites? If so, details of individual samples should be presented as supplementary information.

- Additional methods to select K should be explored (see Janes et al 2017 Molecular Ecology). How were the tested values of K selected? (they differ between RAD-seq and microsatellite analyses).

- Was there a check for clones performed? The authors mention that samples belonged to different genets, but there is no mention of how this was determined. It would be good to show these results.

- The authors present point estimates of population statistic values, such as Fis. With hundreds/thousands of markers these statistics are more informative when shown and discussed as distributions

- The logic behind considering Read 1 and Read 2 of the paired end as replicates is not clear. How are they replicates? This needs to be better justified/explained.

- Lines 327-329. Numbers of fragments are discussed. Please provide the actual numbers (expected vs. observed). Postulate an explanation for discrepancy (see Herrera et al 2015 GBE).

- Genetic diversity is presented in values of pi and also in terms of SNPs/bp. Please present in consistent units.

- Paragraph lines 350-367. What is the main message of this paragraph? It ends without a tying end or conclusion.

- Line 371. What is the predicted nucleotide diversity of A. palmata? Please specify. Also see Arnold et al 2013 Molecular Ecology for a discussion of genetic diversity estimation from RAD-seq data.

Validity of the findings

The data is seemingly robust and the statistical analyses performed are appropriate. However, a greater exploration of the parameters used in the populations analysis in STACKS should be performed. The authors selected conservative parameters regarding the number of individuals per population and the number of populations for a particular locus to be included in the analyses. These parameters have great influence in the number of SNPs that are included in the analyses. The authors correctly identified the relatively low number of SNPs as a weakness in some of their analyses. Performing the population structure and selection analyses in SNPs datasets produced with a variety of parameters is necessary to understand the robustness of the conclusions. Furthermore, the authors should explore additional methods to evaluate population structuring such as PCA, which does not necessarily bins populations into discrete K categories.

---

## Round 0.2 · accepted · Accept

Two reviewers who previously had indicated that the manuscript needed major revision now find the manuscript acceptable for publication. I agree with their assessment. Reviewer 1 remained unconvinced about a couple of points, but I think there is room for legitimate disagreement on these points, and these issues will be worked out by further data collection and analysis in this system and others. Overall, this is a useful contribution to the study of a coral species under serious threat, and makes concrete suggestions for approaches to aid in restoration.

Reviewer 1 ·

Basic reporting

The authors have done a fine job in their revision of this manuscript. The revision clarified the ambiguous methods and results in the originally submitted manuscript. The addition of the supplemental figures for analyses exploring impact of cut-offs for inclusion of loci is useful. I have no further comments for this section.

Experimental design

The experimental design is good and the authors have clarified questions I had with the analysis of the data. I have no further comments for this section.

Validity of the findings

The authors have done what they can do in their revision to address my concerns with the RAD data. Frankly, I am not entirely convinced that the sequence-based data are robust because of the low coverage. I understand that sometimes projects do not go as planned. The authors now include some statements in the manuscript to explain these shortcomings. I have no further comments for this section.

Additional comments

The authors have done a good job with this revision, overall. The section in the Discussion that summarizes the genetic diversity is various animals (beginning on Line 390) still seems unnecessary because these are different types of comparisons (e.g. whole genome compared, small regions of the genome, transcriptomes), thus difficult to compare with any degree of confidence. The authors acknowledge their estimates are potentially not representative for this species (Lines 426-429). I would suggest again that the section "Genes under positive selection" adds little to the manuscript and the manuscript would present better if this were removed. There are few hits to a potential protein and the function would be even further removed.

Reviewer 3 ·

Basic reporting

The authors have satisfactorily addressed the review points.

Experimental design

The authors have satisfactorily addressed the review points.

Validity of the findings

The authors have satisfactorily addressed the review points.

Additional comments

The authors have satisfactorily addressed the review points.